# Symbolic Explanations for Hyperparameter Optimization

Sarah Segel[1]  Helena Graf[1]  Alexander Tornede[1]  Bernd Bischl[2,3]  Marius Lindauer[1]

[1]Institute of Artificial Intelligence, Leibniz University Hannover
[2]Department of Statistics, Ludwig-Maximilians-University Munich
[3]Munich Center for Machine Learning (MCML)

**Abstract**  Hyperparameter optimization (HPO) methods can determine well-performing hyperparameter configurations efficiently but often lack insights and transparency. We propose to apply symbolic regression to meta-data collected with Bayesian optimization (BO) during HPO. In contrast to prior approaches explaining the effects of hyperparameters on model performance, symbolic regression allows for obtaining explicit formulas quantifying the relation between hyperparameter values and model performance. Overall, our approach aims to make the HPO process more explainable and human-centered, addressing the needs of multiple user groups: First, providing insights into the HPO process can support data scientists and machine learning practitioners in their decisions when using and interacting with HPO tools. Second, obtaining explicit formulas and inspecting their properties could help researchers understand the HPO loss landscape better. In an experimental evaluation, we find that naively applying symbolic regression directly to meta-data collected during HPO is affected by the sampling bias introduced by BO. However, the true underlying loss landscape can be approximated by fitting the symbolic regression on the surrogate model trained during BO. By penalizing longer formulas, symbolic regression furthermore allows the user to decide how to balance the accuracy and explainability of the resulting formulas.

## 1 Introduction

As the performance of machine learning (ML) models crucially depends on the choice of suitable hyperparameter configurations (Bischl et al., 2023; Feurer and Hutter, 2019), a plethora of tools focused on finding such configurations has been developed during the last years (Akiba et al., 2019; Awad et al., 2021; Balandat et al., 2020; Falkner et al., 2018; Li et al., 2018; Lindauer et al., 2022). Despite their efficiency in finding a well-performing configuration quickly, these tools are sometimes not adopted in practice - an important reason for this is the lack of insights into the underlying HPO problem generated by the optimization process and, in turn, the returned hyperparameter configuration (Blom et al., 2021). In fact, Godbole et al. (2023) advocate against plainly applying HPO tools in a recent publication and instead argue in favor of an interactive HPO process tailored to gain insights into the problem.

In this work, we take a step to alleviate this situation by moving toward a more human-centered and interpretable HPO process (Moosbauer, Casalicchio, et al., 2022; Moosbauer, Herbinger, et al., 2021) that generates insights into the underlying HPO problem while searching for a well-performing hyperparameter configuration. To this end, we propose to apply symbolic regression to the meta-data collected with Bayesian optimization (BO) during HPO, allowing us to infer a closed-form expression of the relation between hyperparameters and the performance of a configuration. The interpretability of this expression can be controlled via a parsimony hyperparameter trading off (a) the complexity of the expression and (b) how well it describes the actual dependency between hyperparameter values and model performance. We show that naively applying symbolic regression does not yield an expression with high explanation power due to the bias in the meta-data used for training, which is caused by the HPO process' sampling strategy, and we suggest two strategies to mitigate this problem.

For practitioners, our approach (at the slight additional cost of training a symbolic regression model) allows to obtain concrete insights into the HPO loss landscape at hand (Pushak and Hoos, 2022; Schneider et al., 2022) in the form of a formula analytically describing how hyperparameter values influence the performance of a configuration. These insights can be compared and validated against available domain knowledge much more directly than for a black-box surrogate model. Moreover, although we focus on BO-based HPO in this work, the approach can be applied on top of any sampling- and surrogate-based HPO tool, which collects corresponding meta-data. As such, it does not only make HPO more interpretable and insightful for a practitioner, but it is also applicable in a versatile manner, giving it the potential for a large impact on HPO in practice.

In an extensive experimental study, we show that the symbolic expressions indeed faithfully model the HPO loss landscape in terms of a low approximation error while still being of a low enough complexity to be interpretable. In addition, we analyze the trade-off between faithfulness and interpretability based on a sensitivity analysis of the parsimony hyperparameter of our approach.

Altogether, we make the following contributions:

1. We propose an easy workflow offering practitioners the ability to gain insights into the HPO process at little additional cost in the order of minutes.

2. As part of this, we leverage symbolic regression to learn an analytical, closed-form expression to model the dependency between the model performance, i.e., the performance of an HPO configuration, and the values of the corresponding hyperparameters.

3. We show that learning such a symbolic model naively on meta-data collected during the HPO process does not yield a good explanation due to the bias caused by the HPO sampling strategy.

4. We propose a systematic approach to support a practitioner in controlling the trade-off between the symbolic model's faithfulness and interpretability based on a parsimony hyperparameter.

5. Our approach can deal with many hyperparameters by focusing on the most important ones according to functional ANOVA and integrating out the rest using a partial dependence function.

## 2 Related Work

Increased efforts towards explainable HPO approaches have been made by Hutter et al. (2014) and Jin (2022), who calculate hyperparameter importance values analogously to the concept of feature importance, and by Moosbauer, Herbinger, et al. (2020, 2021) and Probst et al. (2019), who leverage partial dependence plots (PDPs) as a posthoc explanation of the HPO process. PDPs (Friedman, 2001) can be used to visualize the marginal effect of one or two features on a model's prediction, i.e., in the context of HPO via BO, the marginal effect of a hyperparameter on the cost predicted by the surrogate model. A significant obstacle to the usage of PDPs for the analysis of the HPO process is posed by the bias introduced through the BO process due to the exploitative sampling of configurations during the search. While this limitation is addressed in a follow-up work by splitting up the hyperparameter space by certainty of the surrogate model as well as modifying the sampling strategy to mitigate the bias in the sampling (Moosbauer, Casalicchio, et al., 2022), the approach is inherently limited to the analysis of one or two hyperparameters at once due to the nature of the PDPs. Very recently, Sass et al. (2022) and Zöller et al. (2022) published first packages that unify a variety of explainability methods in frameworks that allow directly analyzing the optimization process of several automated machine learning tools.

Further, symbolic regression has recently successfully been applied in the context of explainable AutoML. Gijsbers et al. (2021) aim to learn a symbolic formula mapping dataset meta-features to default hyperparameter configurations. While explainability is not the main goal, the approach shows that symbolic regression can be used to find formulas that suggest hyperparameter configurations with competitive or better performance compared to hand-picked or default configurations.

$$s(\alpha, \text{batch size}) = 0.078 \cdot \exp\left(\left(\alpha/\text{batch size}\right)^{\frac{1}{4}}\right)$$

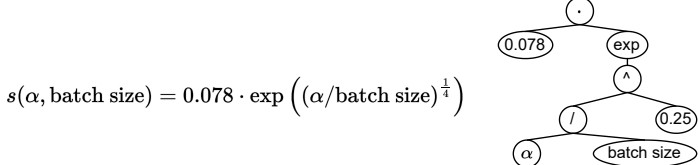

Figure 1: An example of a symbolic expression $s$, visualized as an expression tree. The formula expresses how two hyperparameters typical to neural networks, the regularization factor $\alpha$ and the batch size, influence the predictive performance of the network.

In comparison to our approach, the authors map dataset features to default configurations, whereas we aim to explain the relation between hyperparameters and costs of configurations.

## 3 Background

As a foundation for discussing our approach, aimed at making the HPO process more insightful and explainable, we provide brief introductions of HPO, BO, and symbolic regression.

**Hyperparameter Optimization (HPO).** Carefully choosing the hyperparameters of an ML model for a new task is essential to good model performance (Bischl et al., 2023; Feurer and Hutter, 2019). For a given dataset $\mathcal{D} = \{x_i, y_i\}_{i=1}^n \subset \mathcal{X} \times \mathcal{Y}$, the goal of HPO is to find a hyperparameter configuration $\lambda^* \in \Lambda$ from the configuration space $\Lambda \subseteq \mathbb{R}^d$ that minimizes a given cost function, i.e. $\lambda^* \in \arg\min_{\lambda \in \Lambda} c(\lambda)$. The cost function $c : \Lambda \to \mathbb{R}$ is an unknown black-box function that reflects the validation loss on the given dataset $\mathcal{D}$ of the model instantiated with the given hyperparameter configuration. For ease of notation, we do not make the dependence of $c$ on the dataset $\mathcal{D}$ explicit.

**Bayesian Optimization (BO).** BO is a global optimization procedure that searches for the optimum of a given black-box function by sequentially evaluating the function at certain points (Mockus, 1989). A surrogate model is continuously updated with these points and serves as a basis for an acquisition function that, in turn, suggests the next evaluation point. Due to its sample efficiency and lack of assumptions about the optimized function, it is a popular approach to HPO and has compared favorably to manually tuning hyperparameters or using strategies such as random or grid search (Bergstra et al., 2011; Bischl et al., 2023; Snoek et al., 2012; Turner et al., 2021). When applying BO to HPO, the optimized black-box function is the cost function $c : \Lambda \to \mathbb{R}$, which should be minimized. Usual choices for surrogate models are Gaussian processes (GP) (Rasmussen and Williams, 2006) or, especially in the case of HPO, where the optimized function can have many parameters, a random forest (Breiman, 2001). As a drawback to their usefulness in the optimization process, the methods' complexity dictates that neither gives a clear insight into the relation between the in- and outputs of the black-box function and, thus, the optimization process.

**Symbolic Regression (SR).** In contrast to solely performance-oriented approaches, SR approximates an unknown function from a set of samples $\mathcal{D} = \{x_i, y_i\}_{i=1}^n \subset \mathcal{X} \times \mathcal{Y}$ by constructing a model $s : \mathcal{X} \to \mathcal{Y}$ from mathematical operators, constants, and input variables with the goal to strike a balance between simplicity and accuracy of fit (Augusto and Barbosa, 2000). SR models can be represented as trees of expressions, as illustrated in Figure 1. If their complexity is sufficiently limited, they provide an explainable relationship between in- and outputs through the found expression itself as well as visualizations thereof in terms of plots or tree structures. Finding such an expression is an NP-hard problem; as such, it is often approximately solved with approaches such as genetic programming (Koza, 1994) and recently also methods employing gradient-descent (Alaa and Schaar, 2019; Crabbe et al., 2020; Petersen et al., 2021). Genetic programming refers to a class of evolutionary algorithms, which leverages a tree structure as genome representation. It

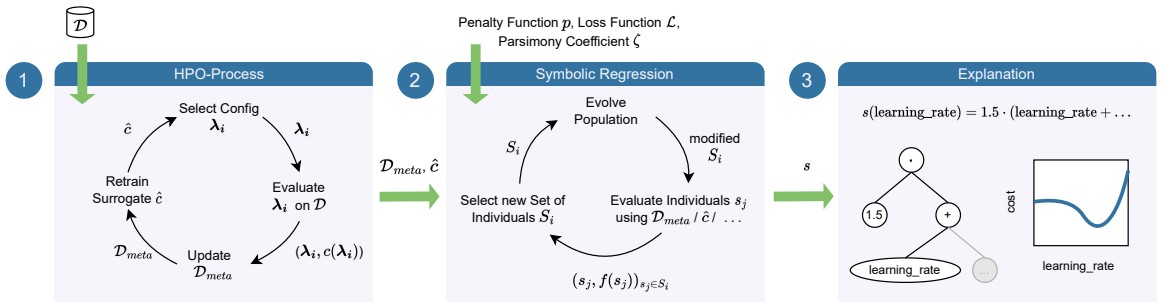

Figure 2: This figure visualizes the overall process of our symbolic HPO explanation approach. The phases correspond to the ones described in Section 4.

iteratively refines a set $S \subset \mathcal{S}$ of individuals $s \in S$ by creating new candidates through genetic operations such as crossover and mutation, evaluating their fitness according to a fitness function $f : \mathcal{S} \to \mathbb{R}$, and subsequently selecting a new set of individuals based on their fitness.

## 4 Learning and Leveraging Symbolic Explanations

Arguably, automating hyperparameter optimization in an efficient way while still generating valuable insights for the practitioner is a challenging task. To move to such a more human-centered HPO process, we propose to learn a numerical expression explaining the relation between the values of optimized hyperparameters and the corresponding performance. The overall process is visualized in Figure 2 and consists of the following steps:

1. Run a BO-based HPO tool and collect (a) the meta-data consisting of the evaluated configurations and their performance and (b) the final surrogate model.

2. Learn a symbolic regression model on either (a) the collected meta-data, or (b) randomly sampled configurations, which are evaluated using the true cost function, or (c) randomly sampled configurations, whose performance is estimated using the Gaussian process.

3. Leverage the symbolic explanation of the underlying HPO loss landscape.

In the following sections, we explain Steps 2 and 3 in more detail, followed by an elaboration on the limitations of our approach.

### 4.1 Learning a Symbolic Hyperparameter Explanation

To learn a symbolic explanation of the relation between the hyperparameters' values and the corresponding configuration's performance, appropriate training data is needed. The most straightforward way is to learn a symbolic model $s : \Lambda \to \mathbb{R}$ from meta-data consisting of $N$ configurations collected during the HPO process, which is of the form

$$\mathcal{D}_{meta} = \left\{ \left( \boldsymbol{\lambda}_i, c(\boldsymbol{\lambda}_i) \right) \right\}_{i=1}^{N} \subset \Lambda \times \mathbb{R} \ . \tag{1}$$

Although there is some work on learning such a symbolic model using gradient-based methods (Alaa and Schaar, 2019; Crabbe et al., 2020; Petersen et al., 2021), we found those methods to be brittle in preliminary experiments, i.e. in some cases we could not recover simple functions, such as polynomial functions, with the approach. Thus, in our experiments we use the classical approach leveraging genetic programming (Koza, 1994).

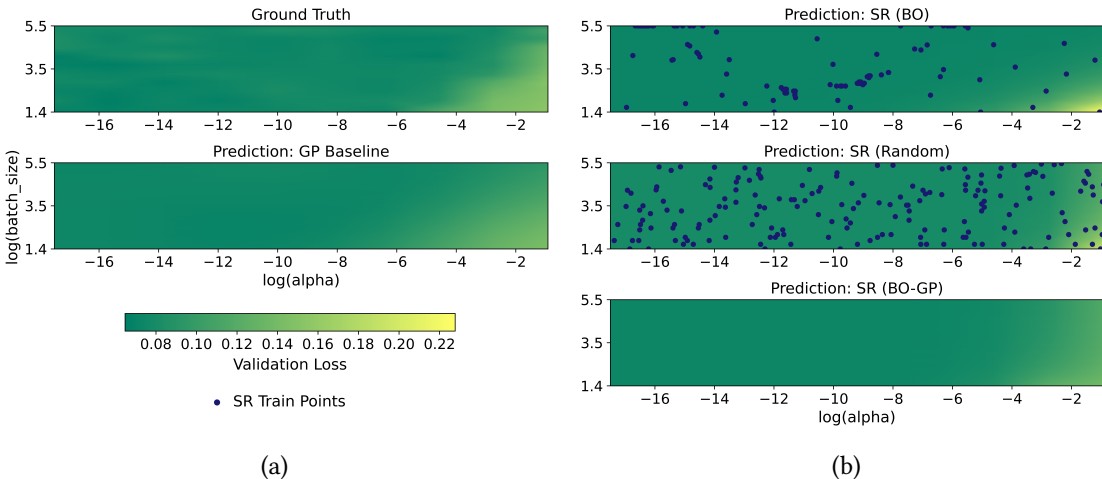

(a)                                                                (b)

Figure 3: This figure displays several representations of the HPO loss landscape defined by tuning the (log) regularization hyperparameter $\alpha$ and the (log) batch size of a neural network on the kc1 dataset with a limit of 200 samples for the HPO process. (a) The top element approximates the ground truth loss landscape by a fine-tuned grid-search over the configuration space. The bottom element visualizes the loss landscape obtained from the actual surrogate model of the HPO process. (b) The top element shows both the configurations sampled by the HPO process and the loss landscape obtained from the symbolic model fitted on the sampled meta-data. The middle element shows randomly sampled configurations and the loss landscape obtained from the symbolic model fitted on them. The bottom element visualizes the loss landscape obtained from the symbolic model fitted on the randomly sampled configurations, whose performance is approximated using the surrogate model of the HPO process.

### 4.1.1 Mitigating Sampling Bias.
As indicated by our experimental evaluation and recently observed by Moosbauer, Herbinger, et al. (2021), models trained on meta-data generated during the search process tend to be biased towards the well-performing regions of the HPO loss landscape. This is caused by the HPO process sampling these points, as it will - in most cases - prefer such well-performing regions, which leads to a $\mathcal{D}_{meta}$ whose points are not even roughly equally distributed across the entire HPO search space. This is visualized in Figure 3b, which shows the configurations sampled by the HPO tool at the top, and randomly drawn configurations in the middle. Correspondingly, directly training on the sampled data can lead to an explanation biased towards those regions.

As Figure 3b (middle) suggests, a solution to this problem is to fit the symbolic model $s$ on randomly sampled configurations instead of the ones sampled by the HPO process to reach a sufficient coverage of the search space and a good prediction performance. However, this comes with the drawback that these randomly sampled points might not have been evaluated during the HPO run, leading to an increased cost for the required additional evaluation of those points. This cost can be circumvented by not evaluating these points, but leveraging the surrogate model $\widehat{c}$ learned during the HPO process in order to predict the performance of these randomly sampled points. Although this offers a potential for error propagation if the surrogate model has a bad fit and / or bias, this strategy tends to work well in practice as our experimental evaluation corroborates.

### 4.1.2 Ensuring Interpretability.
To ensure that the resulting formula is still of an interpretable complexity, we employ a regularization penalty based on the size of the formula. More precisely, we assign a fitness $f : \mathcal{S} \to \mathbb{R}$ to the candidates of the populations to be a linear combination of their predictive performance and the complexity penalty of a formula $s : \Lambda \to \mathbb{R} \in \mathcal{S}$, i.e.,

$$f_c(s) = \mathcal{L}(c, s) + \zeta \cdot p(s) \ . \tag{2}$$

Here, $\mathcal{L}(c, s)$ returns the loss between the (approximated) performance of the sampled configurations according to a cost function $c$ and the predicted cost according to the symbolic model $s$ on the above mentioned training dataset $\mathcal{D}_{meta}$ (Equation 1), $p(s)$ returns the complexity penalty for formula $s$ and $\zeta \in \mathbb{R}$ is a parsimony hyperparameter of our approach. For the third instantiation of our approach, we use the surrogate $\widehat{c}$ instead of $c$ to approximate the performance of the sampled configurations as outlined at the beginning of Section 4. For this work, we chose the root-mean-square error for $\mathcal{L}$ and the length, i.e., number of operations of $s$, as the penalty $p$.

In order to achieve a desired complexity, the parsimony hyperparameter $\zeta$ has to be set accordingly. Due to the potential difference between the scale of the loss $\mathcal{L}$ and the penalty function $p$, doing so is not a straightforward task. Similar problems are not unknown in the meta-algorithmics community, e.g., in combined ranking and regression approaches for algorithm selection (Fehring et al., 2022; Hanselle et al., 2020). As a solution, we suggest to leverage the elbow heuristic (Thorndike, 1953), which is well-known in clustering. The underlying idea is to fit the symbolic regression multiple times with different parsimony values as visualized in Figure 5, starting with a high value and decreasing it until the increase in performance no longer outweighs the increasing complexity and thus decreasing explainability. More details on this idea can be found in Appendix A.

### 4.1.3 Subsets of Hyperparameters.
The set of hyperparameters to be included in the explanation and the one being optimized has to be the same. Otherwise, the construction of a meta-data training dataset for the SR is problematic due to noise imposed onto the performance by the hyperparameters unconsidered for the explanation. To alleviate this, we propose to extend our workflow: After running the HPO in the first step, functional ANOVA (Hutter et al., 2014) can be applied to calculate the importance of all hyperparameters. We propose to select the hyperparameters with the highest importance to explain them. We can leverage the partial dependence (PD) function, as done by Moosbauer, Herbinger, et al. (2021), which expresses the expected performance of a configuration independent of the irrelevant hyperparameters by integrating them out. After computing the partial dependence for the corresponding set of hyperparameters, steps two and three of our approach can be applied to obtain an explanation with respect to the most important hyperparameters.

## 4.2 Leveraging a Symbolic Hyperparameter Explanation

Once a symbolic regression model, as outlined in Section 4.1, is learned, we can quantify the relation between the values of the hyperparameters and the performance of the corresponding configuration in the form of a closed-form analytic expression, i.e., a formula. Below, we outline some exemplary use cases for such a symbolic hyperparameter explanation.

### 4.2.1 Characterizing the HPO Loss Landscape.
While there has been work on characterizing the loss landscape of HPO problems (Pushak and Hoos, 2022; Schneider et al., 2022), the focus has mostly been on indirect characterization ways by computing certain statistics on sampled configurations and their performance. The symbolic model allows for a much more concrete characterization of the landscape in the form of a formula. This also gives an easy way to plot the landscape as Figure 3a and Figure 3b demonstrate. In Figure 3a, the top element approximates the ground truth landscape based on a high-resolution grid-search, the bottom element shows the loss landscape as predicted by the original HPO surrogate model whereas, in Figure 3b, the bottom element shows the loss landscape as predicted by the symbolic explanation model.

### 4.2.2 Theoretical Analysis of HPO Loss Landscapes.
A concrete closed-form expression quantifying the relation between the values of hyperparameters and the performance of the corresponding configuration has the potential to pave the way for a theoretical analysis of HPO loss landscapes. Under the assumption that the formula is at least roughly correct, theoreticians might be able to analyze the concrete interaction effects of hyperparameters on the performance, potentially across several datasets, and thus might gain insights into why certain algorithms perform very well on some datasets and worse on others.

### 4.3 Limitations

As of now, our approach can only be used if the HPO configuration space solely consists of numerical and no categorical or other types of hyperparameters. This limitation arises as the formulas assume the hyperparameters to be real numbers. Moreover, ensuring interpretability via the parsimony coefficient controlling the penalty is a very simple solution to the actual underlying multi-objective optimization problem. We discuss these limitations in more detail in Appendix C.

## 5 Experimental Evaluation

In this section, we show that symbolic models can provide simple explanations of the HPO loss landscape while faithfully capturing its dynamics. Leveraging a subset of problems from HPOBench (Eggensperger et al., 2021), a collection of benchmark problems for HPO, we apply SR to learn the dependency between hyperparameter values and performance for a large range of models, hyperparameters, and datasets. As the cost associated with a hyperparameter configuration, we consider the validation error rate evaluated via hold-out testing, using 0.33% of each dataset for validation. We use the BO-powered HPO tool SMAC (Lindauer et al., 2022) as a basis for the evaluation (Phase 1 in the description in Section 4) and compare three ways of training a symbolic explanation based on the HPO run as described in Section 4. SR (BO) refers to a symbolic regression fitted based on the meta-data collected by the HPO process, SR (Random) refers to a training based on randomly sampled configurations, which are evaluated, whereas SR (BO-GP) refers to a training based on randomly sampled points, whose performance is approximated using the posterior mean of the surrogate model of the HPO process, in this case a Gaussian process (GP). Technical details on the experiments, such as the number of seeds, the hardware for execution, and the used resources, can be found in Appendix D. The code for running the experiments is provided online[1]. In the following sections, we analyze the accuracy of the symbolic explanation (Section 5.1) under varying HPO sample sizes and the trade-off between the faithfulness and explainability (Section 5.2).

### 5.1 Faithfulness of Symbolic Explanations

In Table 1, the root-mean-square error (RMSE) between the cost values predicted by the SR and the true cost values averaged over 100 test configurations is shown for the three meta-data generation procedures described in Section 4.1. The RMSE between the cost predicted by the GP and the true cost is shown as a baseline. As we expect due to sampling bias, when fitted on the BO samples directly, the RMSE is highest for most models and datasets, while fitting on random samples results in the lowest error in most cases. Leveraging the GP to obtain the cost predictions for the random samples results in error values close to those we obtain when using the true cost values and an even lower average error. This suggests that this approach is suitable to obtain a faithful explanation without evaluating additional hyperparameter configurations as SR (Random) does. Furthermore, we compare the SR against a linear regression, showing that for some model and dataset combinations the difference in performance is minor, while for others it is substantial.

In addition, we study how the error of the symbolic regression and the GP baseline depends on the number of samples it is fit on, i.e., how faithful the symbolic explanation is under varying sample sizes. Figure 4 displays the results for two models on two datasets. Our results show that the error decreases slightly between 20 and 60 samples. However, adding more samples does not have a large effect on the error, suggesting that a small number of samples is sufficient for the studied models and datasets to apply symbolic regression as explainer. For more results, see Appendix E.

### 5.2 Faithfulness vs. Interpretability

By penalizing longer formulas, SR allows to balance the accuracy and interpretability of the resulting formulas via a parsimony hyperparameter $\zeta$. Figure 5 shows the RMSE between the cost predicted

---

[1] https://github.com/automl/symbolic-explanations

| Model | Hyperparameters | Dataset | SR (BO) | SR (Random) | SR (BO-GP) | GP Baseline | Linear Regression |
|---|---|---|---|---|---|---|---|
| Logistic Regression | alpha, eta0 | blood-transfusion | 0.015 ± 0.006 | **0.010 ± 0.002** | 0.016 ± 0.005 | 0.019 ± 0.009 | 0.015 ± 0.006 |
| | | vehicle | 0.045 ± 0.006 | **0.035 ± 0.003** | 0.041 ± 0.004 | 0.026 ± 0.004 | 0.062 ± 0.002 |
| | | Australian | 0.361 ± 1.275 | **0.023 ± 0.007** | 0.024 ± 0.005 | 0.024 ± 0.006 | 0.025 ± 0.003 |
| | | car | 0.029 ± 0.007 | **0.025 ± 0.002** | 0.026 ± 0.002 | 0.018 ± 0.002 | 0.052 ± 0.001 |
| | | phoneme | 0.012 ± 0.003 | **0.010 ± 0.003** | 0.011 ± 0.001 | 0.011 ± 0.003 | 0.010 ± 0.001 |
| | | segment | 0.062 ± 0.036 | **0.028 ± 0.016** | 0.033 ± 0.005 | 0.035 ± 0.005 | 0.053 ± 0.002 |
| | | credit-g | 0.028 ± 0.006 | **0.024 ± 0.004** | 0.028 ± 0.002 | 0.026 ± 0.001 | 0.026 ± 0.001 |
| | | kc1 | 0.017 ± 0.004 | 0.020 ± 0.004 | **0.016 ± 0.003** | 0.015 ± 0.001 | 0.015 ± 0.002 |
| Support Vector Machine | C, gamma | blood-transfusion | 0.036 ± 0.015 | **0.032 ± 0.005** | 0.034 ± 0.002 | 0.008 ± 0.001 | 0.031 ± 0.000 |
| | | vehicle | 0.133 ± 0.025 | **0.094 ± 0.008** | 0.097 ± 0.009 | 0.071 ± 0.014 | 0.234 ± 0.001 |
| | | Australian | 0.078 ± 0.012 | **0.062 ± 0.005** | 0.069 ± 0.010 | 0.061 ± 0.011 | 0.139 ± 0.000 |
| | | car | 0.070 ± 0.008 | 0.053 ± 0.003 | **0.052 ± 0.002** | 0.032 ± 0.010 | 0.103 ± 0.000 |
| | | phoneme | 0.067 ± 0.027 | **0.044 ± 0.008** | 0.047 ± 0.006 | 0.019 ± 0.002 | 0.081 ± 0.000 |
| | | segment | 0.178 ± 0.040 | **0.128 ± 0.019** | 0.133 ± 0.013 | 0.100 ± 0.011 | 0.282 ± 0.001 |
| | | credit-g | 0.041 ± 0.008 | **0.038 ± 0.007** | 0.039 ± 0.005 | 0.020 ± 0.003 | 0.080 ± 0.000 |
| | | kc1 | 0.034 ± 0.015 | 0.029 ± 0.006 | **0.024 ± 0.006** | 0.007 ± 0.001 | 0.034 ± 0.000 |
| Random Forest | max_depth, max_features | blood-transfusion | 0.016 ± 0.004 | **0.015 ± 0.004** | 0.017 ± 0.000 | 0.008 ± 0.001 | 0.014 ± 0.000 |
| | | vehicle | 0.034 ± 0.046 | 0.021 ± 0.008 | **0.020 ± 0.004** | 0.018 ± 0.003 | 0.109 ± 0.004 |
| | | Australian | 0.024 ± 0.009 | 0.022 ± 0.008 | **0.018 ± 0.002** | 0.016 ± 0.003 | 0.033 ± 0.002 |
| | | car | 0.035 ± 0.005 | **0.030 ± 0.006** | 0.031 ± 0.005 | 0.024 ± 0.002 | 0.071 ± 0.003 |
| | | phoneme | 0.015 ± 0.002 | 0.016 ± 0.002 | **0.015 ± 0.002** | 0.008 ± 0.002 | 0.052 ± 0.002 |
| | | segment | 0.040 ± 0.007 | **0.035 ± 0.004** | 0.039 ± 0.005 | 0.029 ± 0.006 | 0.147 ± 0.003 |
| | | credit-g | 0.025 ± 0.012 | 0.022 ± 0.006 | **0.021 ± 0.003** | 0.012 ± 0.002 | 0.036 ± 0.002 |
| | | kc1 | **0.015 ± 0.006** | 0.015 ± 0.005 | 0.019 ± 0.001 | 0.005 ± 0.002 | 0.013 ± 0.000 |
| XGBoost | colsample_bytree, eta | blood-transfusion | 0.018 ± 0.002 | **0.017 ± 0.001** | 0.017 ± 0.002 | 0.014 ± 0.001 | 0.016 ± 0.001 |
| | | vehicle | **0.009 ± 0.001** | 0.010 ± 0.001 | 0.009 ± 0.001 | 0.009 ± 0.001 | 0.009 ± 0.001 |
| | | Australian | **0.011 ± 0.000** | 0.011 ± 0.000 | 0.011 ± 0.000 | 0.010 ± 0.001 | 0.010 ± 0.000 |
| | | car | 0.031 ± 0.015 | **0.018 ± 0.004** | 0.020 ± 0.003 | 0.016 ± 0.005 | 0.045 ± 0.000 |
| | | phoneme | 0.013 ± 0.000 | **0.012 ± 0.000** | 0.013 ± 0.000 | 0.008 ± 0.001 | 0.013 ± 0.000 |
| | | segment | **0.003 ± 0.000** | 0.003 ± 0.000 | 0.003 ± 0.000 | 0.003 ± 0.001 | 0.003 ± 0.000 |
| | | credit-g | 0.011 ± 0.001 | **0.011 ± 0.000** | 0.011 ± 0.000 | 0.011 ± 0.001 | 0.011 ± 0.000 |
| | | kc1 | 0.008 ± 0.000 | 0.008 ± 0.000 | **0.008 ± 0.000** | 0.006 ± 0.001 | 0.007 ± 0.000 |
| Neural Network | alpha, batch_size | blood-transfusion | 0.016 ± 0.002 | **0.013 ± 0.001** | 0.014 ± 0.002 | 0.011 ± 0.001 | 0.013 ± 0.000 |
| | | vehicle | 0.024 ± 0.014 | **0.018 ± 0.002** | 0.029 ± 0.008 | 0.030 ± 0.006 | 0.041 ± 0.002 |
| | | Australian | 0.014 ± 0.003 | **0.012 ± 0.003** | 0.014 ± 0.003 | 0.014 ± 0.002 | 0.014 ± 0.001 |
| | | car | 0.010 ± 0.002 | **0.008 ± 0.001** | 0.010 ± 0.002 | 0.007 ± 0.003 | 0.021 ± 0.000 |
| | | phoneme | 0.026 ± 0.010 | **0.012 ± 0.003** | 0.014 ± 0.004 | 0.010 ± 0.003 | 0.027 ± 0.001 |
| | | segment | 0.015 ± 0.008 | **0.012 ± 0.006** | 0.022 ± 0.023 | 0.013 ± 0.004 | 0.025 ± 0.001 |
| | | credit-g | **0.017 ± 0.006** | 0.214 ± 0.722 | 0.019 ± 0.006 | 0.020 ± 0.007 | 0.025 ± 0.001 |
| | | kc1 | 0.015 ± 0.004 | 0.014 ± 0.004 | **0.012 ± 0.002** | 0.007 ± 0.001 | 0.014 ± 0.000 |
| Average | | | 0.0412 | 0.0306 | **0.0274** | 0.0200 | 0.0503 |

Table 1: We show the RMSE between the cost predicted by the SR and the true cost. The SR is fitted either on samples collected by BO (BO), random samples with the true cost (Random) or random samples evaluated with the GP (BO-GP). For each row, the best/second best RMSE among the SR models is boldfaced/underlined. The surrogate model's RMSE (GP Baseline) is shown for comparison, as well as the RMSE of a linear regression model fitted on the same samples as SR (GP-BO). The SR is fitted on 140 samples with $\zeta = 0.0001$. For GP-BO, it is fitted on 400 random samples evaluated with the GP obtained after running BO for 140 samples.

by the SR and the true cost, as well as the complexity expressed in terms of the number of operations in the resulting formulas for multiple values of $\zeta$. As expected, for all shown models and datasets, the error decreases with increasing complexity. We suggest (cf. Section 4.1.2) using the elbow heuristic to select the appropriate value of $\zeta$. The results imply that this strategy is indeed justified.

Two formulas as possible hypotheses obtained by our approach are shown below. They describe the dependency between the regularization hyperparameter $\alpha$ and the batch size of a neural network on the kc1 dataset, with the SR fitted on 400 random samples evaluated with the GP obtained after collecting 200 samples with BO with different seeds for the SR. While the formulas are composed of different functions, it must be considered that they only describe the relationship within the bounds defined for the hyperparameters. Within these bounds, the relationship described by them is quite similar. Notably, the second formula does not depend on the batch size, indicating that $\alpha$ is more important. Figure 6a shows the hyperparameter importance values, calculated using functional ANOVA (Hutter et al., 2014), confirming that $\alpha$ is indeed more important than the batch size. This is in line with recent findings (Godbole et al., 2023).

$$s(\alpha, \text{batch size}) = 0.078 \cdot \exp\left((\alpha/\text{batch size})^{\frac{1}{4}}\right) \quad s(\alpha, \text{batch size}) = 0.104 \cdot \sqrt{|\sin(\alpha)|} + 0.07$$

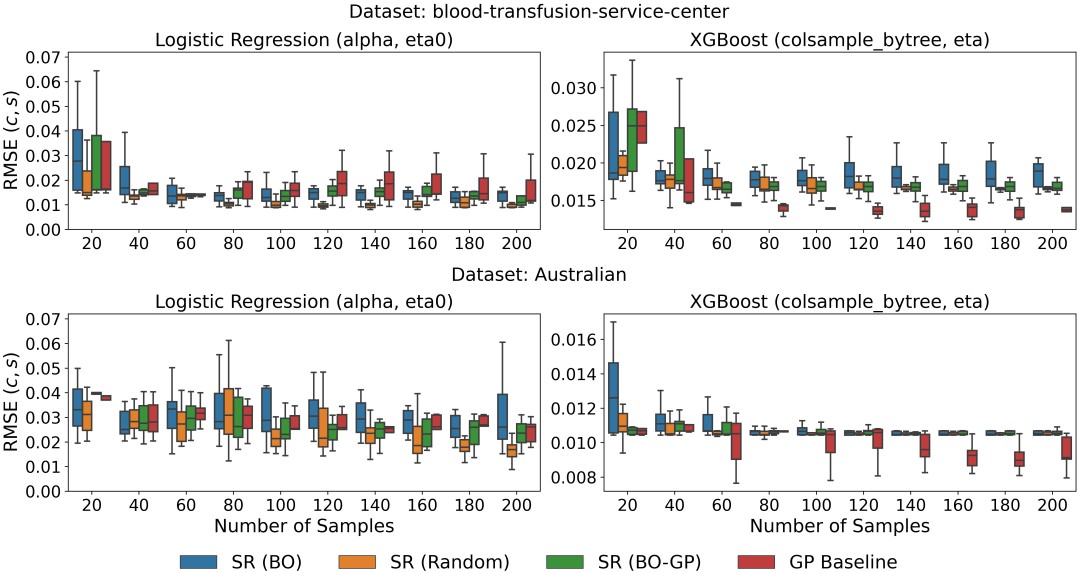

Figure 4: RMSE between the cost predicted by the SR and the true cost for different numbers of samples. The SR is fitted on either samples collected by BO (BO), random samples with their true cost (Random), or random samples evaluated with the GP (BO-GP). The RMSE of the GP (GP Baseline) is shown for comparison. The SR is fitted on the number of samples shown, with parsimony coefficient 0.0001. For GP-BO, it is fitted on 400 random samples evaluated with the GP obtained after collecting the shown number of samples with BO.

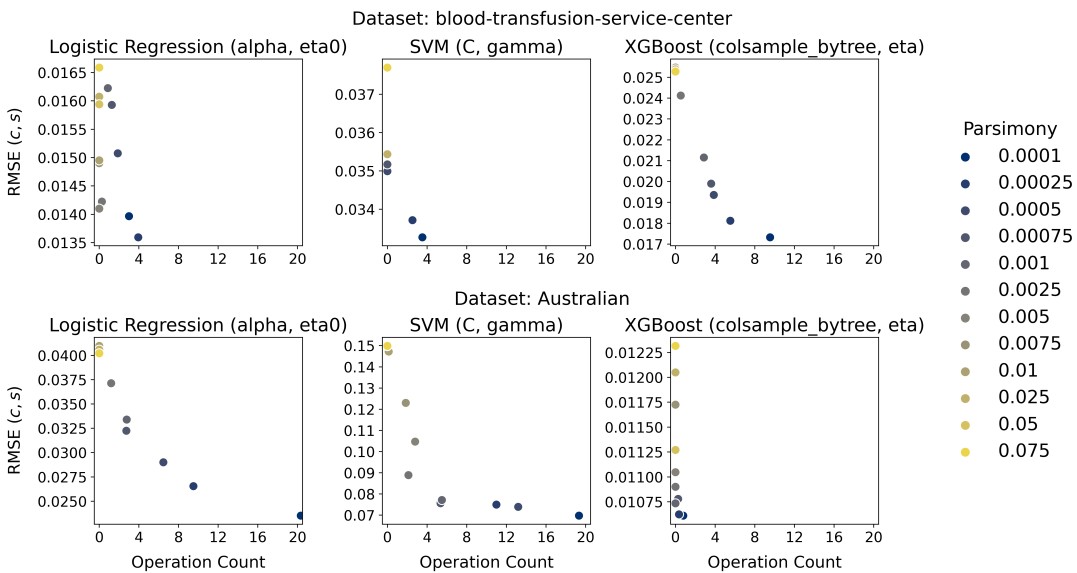

Figure 5: RMSE between the cost predicted by the SR and the true cost for different values of the parsimony coefficient $\zeta$. The symbolic regression is fitted on 400 random samples evaluated with the Gaussian process obtained after collecting 200 samples with BO.

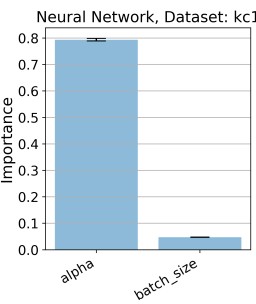

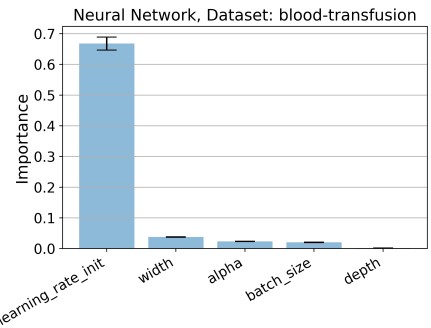

(a) Hyperparameter importance on kc1.      (b) Hyperparameter importance on blood-transfusion.

Figure 6: Importance of several hyperparameters of a neural network calculated by functional ANOVA.

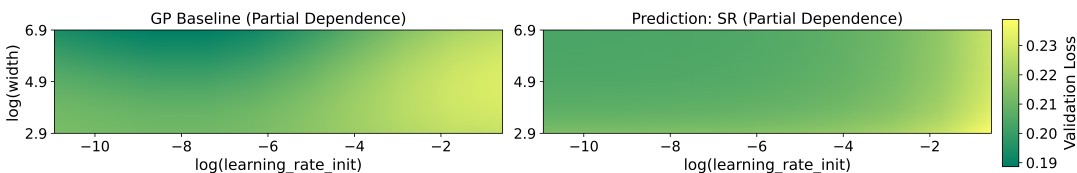

Figure 7: Result of optimizing the learning rate, width, regularization hyperparameter $\alpha$, batch size, and depth of a neural network on the blood-transfusion dataset for 100 samples. Left: PD function for the learning rate and width. Right: Loss landscape obtained from the SR fitted on randomly sampled configurations, whose performance is approximated by the PD function.

Finally, we provide an example on how to obtain formulas for subsets of hyperparameters as described in Section 4.1.3. After optimizing multiple hyperparameters of a neural network, we calculate their importances using functional ANOVA, shown in Figure 6b. Leveraging the GP surrogate model, we then calculate the PD for the two hyperparameters with the highest importance scores and fit the SR. Figure 7 illustrates the PD function and the function obtained from the SR.

## 6 Conclusion and Future Work

In this work, we took a step toward a more human-centered HPO process by proposing an easy workflow for practitioners targeted at gaining insights about the underlying HPO problem. To this end, we leverage symbolic regression to learn an interpretable, analytic closed-form expression of the relation between hyperparameter values and their performance. We show that naively learning such a symbolic model on meta-data collected during the HPO process suffers from bias issues and suggest solving this solution by learning on randomly sampled configurations whose performance is approximated by the HPO process' surrogate model. In an experimental evaluation, we show that a trade-off between the interpretability and faithfulness of the symbolic model can be achieved with a systematic approach for setting a parsimony hyperparameter.

There are several interesting directions toward future work. As most modern HPO tools leverage multi-fidelity techniques to improve efficiency, we deem it worthwhile to extend our approaches to handle meta-data from several fidelities in a suitable manner. Furthermore, it is conceivable to incorporate the symbolic regression model as a surrogate model into BO-based HPO tools. Here, the main challenge is to find a suitable way to model the uncertainty of the symbolic model, which is required by many acquisition functions in practice. Last, one could make use of conformal prediction (Angelopoulos and Bates, 2021) for quantifying the uncertainty of the different expressions obtained by the symbolic regression.

## 7 Broader Impact Statement

HPO and thus also the approach presented in this paper can be used in virtually any ML applications and, in principle, even for most general configuration problems, e.g., in mechanical engineering (Gevers et al., 2022; Kotthoff et al., 2022). Correspondingly, it can have a positive or negative impact on society based on the application it is used for. If it is used with malicious intent or within a socially critical application such as face recognition, it can lead to giving practitioners more insights, which might allow for increasing the performance and, thus, the potential harmfulness of the corresponding tool. However, at the same time, when used within applications with a positive impact on society, such as most medical applications (Imrie et al., 2022), similar improvements are conceivable and conversely, a positive impact on society.

**Acknowledgements**. Funded by the European Union (ERC, "ixAutoML", grant no.101041029). Views and opinions expressed are however those of the author(s) only and do not necessarily reflect those of the European Union or the European Research Council Executive Agency. Neither the European Union nor the granting authority can be held responsible for them.

The authors gratefully acknowledge the computing time provided to them on the high-performance computers Noctua2 at the NHR Center PC2 under the project hpc-prf-intexml. These are funded by the Federal Ministry of Education and Research and the state governments participating on the basis of the resolutions of the GWK for the national high performance computing at universities (www.nhr-verein.de/unsere-partner).

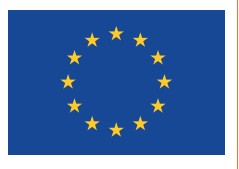
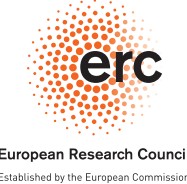

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

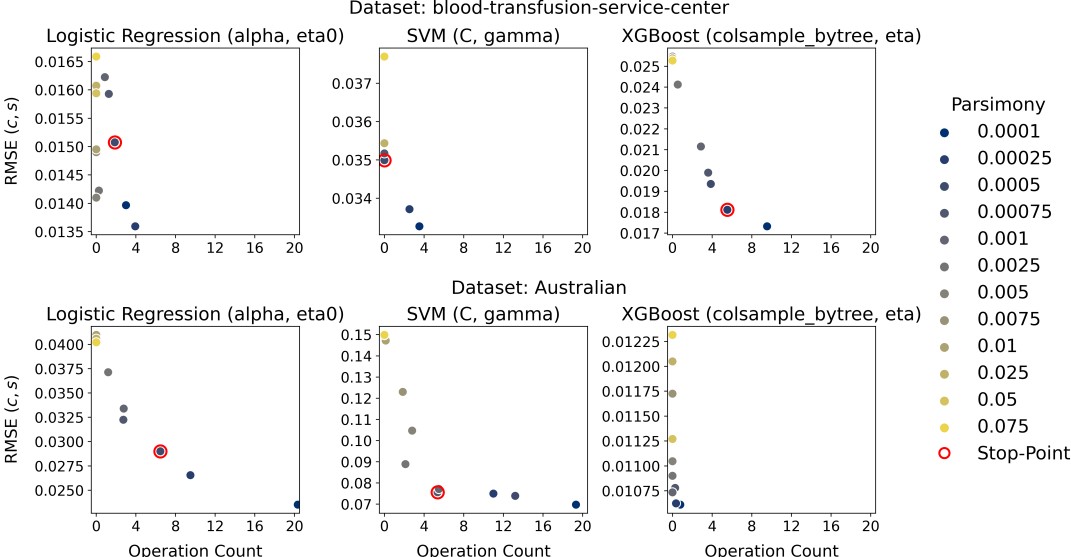

Figure 8: RMSE between the cost predicted by the SR and the true cost for different values of the parsimony coefficient $\zeta$. The symbolic regression is fitted on 400 random samples evaluated with the Gaussian process obtained after collecting 200 samples with BO. The red circle indicates the stop-point, i.e., the value of the parsimony coefficient at which we would stop increasing the parsimony coefficient further according to our heuristic.

## A  Applying the Elbow Heuristic

We propose the use of the elbow heuristic as a systematic approach to balancing the interpretability and faithfulness of the symbolic model via a parsimony parameter. The method suggests starting with a low complexity (and, in turn, usually relatively lower performance), and decreasing the parsimony parameter until the benefits of increased performance no longer outweigh the increased complexity of the resulting formula. One conceivable way a practitioner could put this into practice is by setting thresholds for gain of performance/loss of interpretability. For example, in Figure 8, we stop decreasing the parsimony parameter once the next value of the parsimony parameter would increase the operation count by at least two while decreasing the RMSE by less than 10%. Another option would be to define a relative trade-off point between accuracy and complexity changes where one would stop (e.g., stopping once the increase in complexity is larger than the relative gain of performance).

## B  On the Operation Count for Symbolic Expressions

As a measure for the complexity of a symbolic expression, we use the operation count in the formula as computed by the *SymPy* library. Operations are all functions with an arity $> 0$, i.e. all nodes in an expression tree except leaf nodes. For example, the expression $s(\alpha, \text{batch size}) = 0.078 \cdot \exp\left((\alpha/\text{batch size})^{1/4}\right)$ depicted in Figure 1 has 4 operations. An expression with 0 operations is a constant or a variable.

## C  Limitations

In its current state, our approach has several limitations, which we discuss here together with possible remedies:

**Numerical Hyperparameters**. As of now, our approach can only be used if the HPO configuration space consists solely of numerical and no categorical or other types of hyperparameters. This limitation arises from the fact that hyperparameters are treated as variables in the symbolic regression, whose values must be able to be plugged in correspondingly. As usual in such cases, any other type of hyperparameter can be supported by providing a suitable mapping of the space to a numerical space, such as a one-hot encoding for categorical hyperparameters. However, it is unclear how well this works in practice in this case.

**Multi-Objective Optimization**. As noted in Section 4.1.2, we ensure interpretability by penalizing very complex formulas through a penalty term in their fitness value. This is an instantiation of scalarization (Deb, 2013) for solving multi-objective optimization problems (Deb, 2013). Since we actually want to optimize for both accuracy/faithfulness of the explainer, i.e., the symbolic regression model, and the complexity of the corresponding formula, a more sophisticated approach would be to employ Pareto-based multi-objective approaches. This would also improve the usability for the practitioner as they would not need to specify the penalty in advance but could pick a formula from the Pareto front at the end of the optimization, which offers a compromise between accuracy and interpretability to their taste.

**Subsets of Hyperparameters**. Our approach can deal with many hyperparameters by focusing on the most important ones according to functional ANOVA and integrating out the rest using a partial dependence (PD) function, as discussed in Section 4.1.3. As a result of including functional ANOVA and PD, the extended approach may be subject to their limitations, in particular:

1. Functional ANOVA and PD can yield misleading results when dealing with correlated features due to extrapolation to unlikely combinations of feature values, i.e., hyperparameter values, in our case (Hooker, 2007). However, in contrast to the standard setting of interpreting machine learning models, there is no inherent data distribution on the hyperparameter configuration space in the context of HPO. Thus, this limitation is of no concern for the HPO setting.

2. When interaction effects are present, interpreting the main effects obtained by functional ANOVA can lead to inaccurate conclusions, as they capture the effect of varying a hyperparameter averaging across all instantiations of all other hyperparameters (Hutter et al., 2014). Similarly, PD can be misleading when interaction effects are present, capturing only the average marginal effects (Molnar et al., 2022). However, recent findings by Pushak and Hoos (2022) suggest that there are few strong interactions in HPO problems, which makes these limitations less of a concern in practice.

## D  Detailed Experimental Setup

For all experiments, we used 5 seeds to obtain the training data points, i.e., configurations for the symbolic regression and the corresponding performances. Furthermore, we repeated learning the symbolic expression three times with different seeds. All results are averaged, and error bars as well as standard deviations are computed across the $5 \times 3 = 15$ repetitions. Experiments are run on cluster nodes equipped with two AMD Milan 7763 with 2×64 cores@2.45 GHz and were limited to 64 CPUs and 64GB of RAM. In the interest of GreenAutoML (Tornede et al., 2021), we aim to be transparent regarding the computational resources required for our experiments, which consumed about 700.000 CPU hours in total. As the cluster is exclusively powered by renewable energy, the experiments did not cause any $CO_2$ equivalents when considering only the energy required to run them.

**Bayesian Optimization**. We use the HPO tool SMAC (Lindauer et al., 2022) for running BO in our experiments. We use the BlackBoxFacade with the default settings, except for the config selector,

where we set retrain_after=1 to update the surrogate model each time when returning a new hyperparameter configuration.

**Symbolic Regression**. To learn the symbolic regression models we use *gplearn*[2], a library to perform genetic programming based symbolic regression. We fit the Symbolic Regressor with a population size of 5000, evolving over 20 generations and optimizing the RMSE. The parsimony coefficient is set to 0.0001 in our experiments except for the parsimony variation study. We allow the functions addition, subtraction, multiplication, division, square root, logarithm, exponential, sine, cosine, and absolute value when building and evolving the formulas. For all other hyperparameters, we use the default values. The formulas are simplified with the *SymPy*[3] tool.

## E  Detailed Results

Extending the results presented in Section 5.1, Figure 9, 10, and 11 display the error of the symbolic regression and the GP baseline for varying sample sizes for different models, hyperparameters, and datasets.

---

[2]https://gplearn.readthedocs.io/en/stable/intro.html
[3]https://www.sympy.org/en/index.html

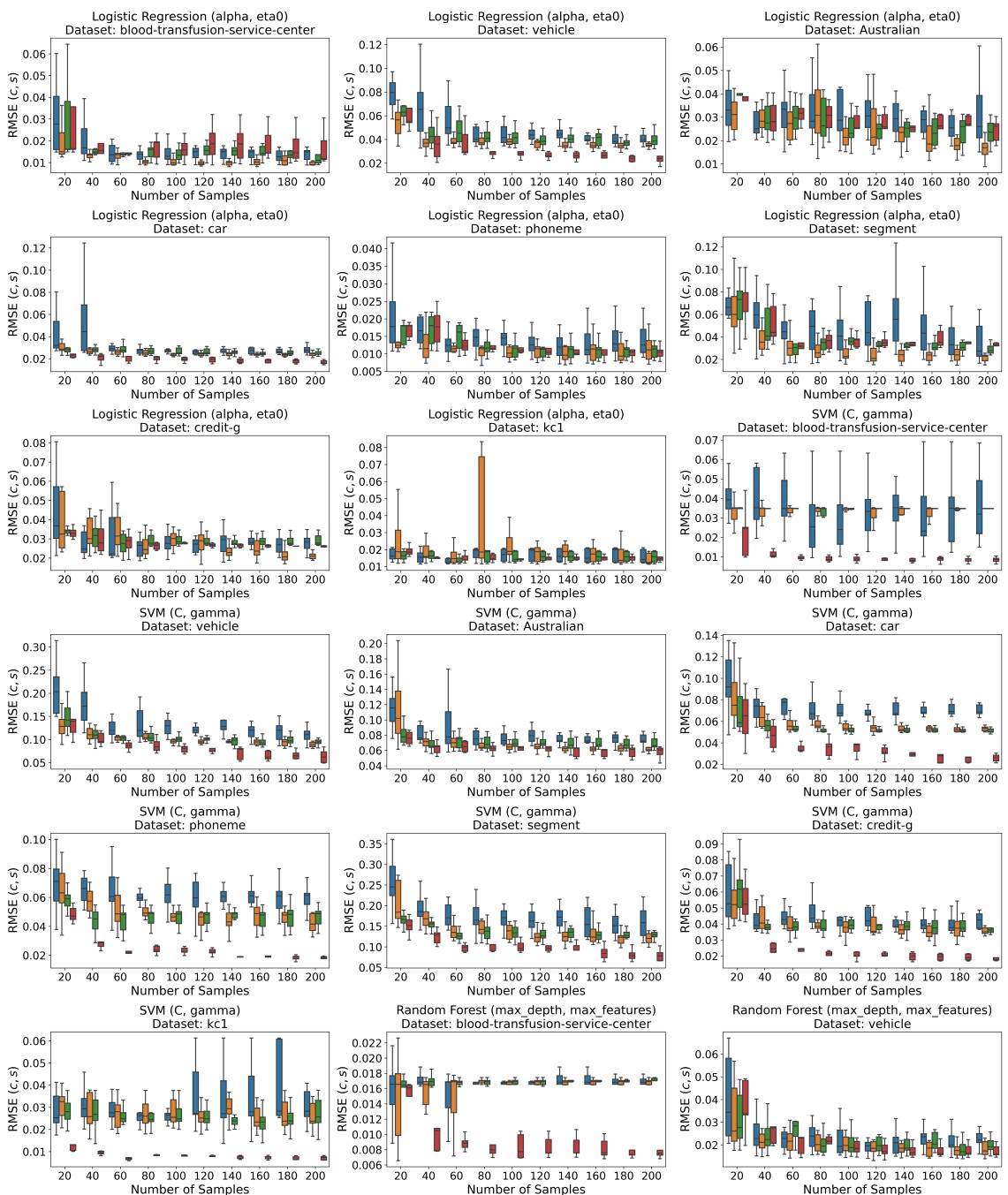

Figure 9: RMSE between the cost predicted by the SR and the true cost for different numbers of samples. The symbolic regression is fitted on either samples collected by BO (BO), random samples with their true cost (Random), or random samples evaluated with the GP (BO-GP). The RMSE of the GP (GP Baseline) is shown for comparison. The symbolic regression is fitted on the number of samples shown, with parsimony coefficient 0.0001. For GP-BO, it is fitted on 400 random samples evaluated with the GP obtained after collecting the shown number of samples with BO.

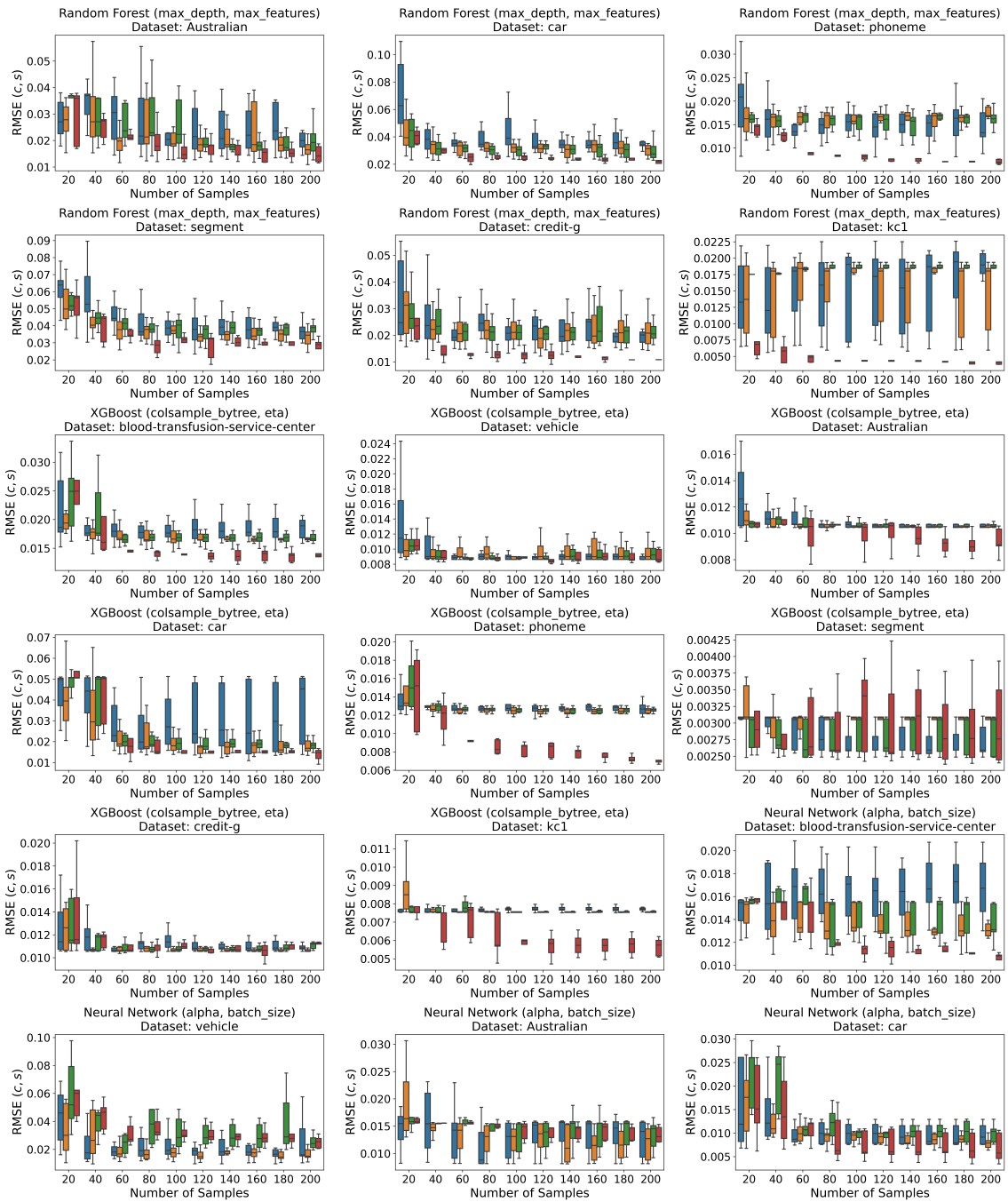

Figure 10: RMSE between the cost predicted by the SR and the true cost for different numbers of samples. The symbolic regression is fitted on either samples collected by BO (BO), random samples with their true cost (Random), or random samples evaluated with the GP (BO-GP). The RMSE of the GP (GP Baseline) is shown for comparison. The symbolic regression is fitted on the number of samples shown, with parsimony coefficient 0.0001. For GP-BO, it is fitted on 400 random samples evaluated with the GP obtained after collecting the shown number of samples with BO.

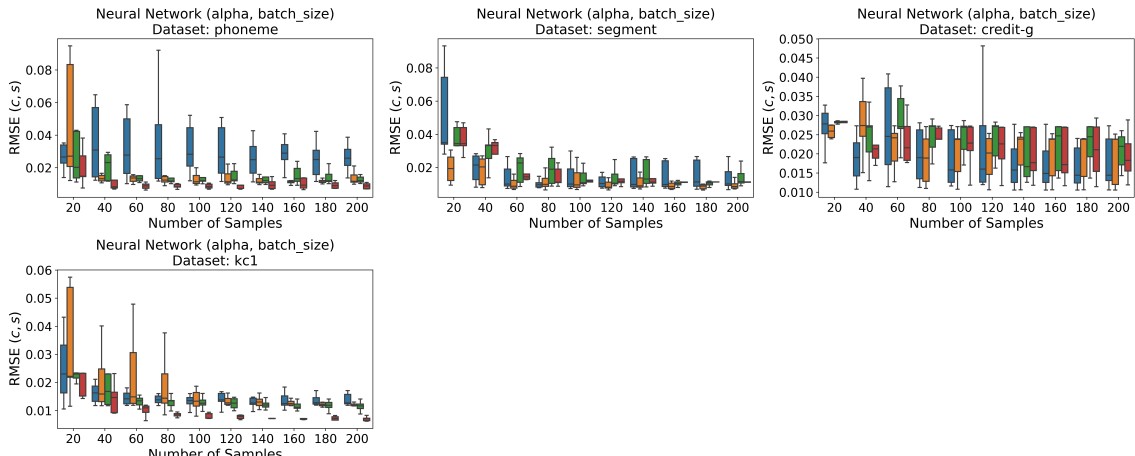

Figure 11: RMSE between the cost predicted by the SR and the true cost for different numbers of samples. The symbolic regression is fitted on either samples collected by BO (BO), random samples with their true cost (Random), or random samples evaluated with the GP (BO-GP). The RMSE of the GP (GP Baseline) is shown for comparison. The symbolic regression is fitted on the number of samples shown, with parsimony coefficient 0.0001. For GP-BO, it is fitted on 400 random samples evaluated with the GP obtained after collecting the shown number of samples with BO.

