# OpenReview forum: "Symbolic Explanations for Hyperparameter Optimization"
_automl.cc/AutoML/2023/Conference — AutoML 2023 MainTrack_

### Official Review · Reviewer_PYEc · 2023-04-10

**Potential Impact On The Field Of Automl Rating:** 3
**Technical Quality And Correctness Rating:** 3
**Clarity Rating:** 3

**Summary Of Contributions:**

This paper introduces a new approach to make the HPO process more explainable via applying symbolic regression to the metadata collected with BO. To this end, the authors develop the inference of a closed-form expression that captures the hyperparameter values and its performance with controllable tradeoffs. Empirical experiments show that the proposed method and the systematic approach is able to faithfully model the HPO loss landscape with the tradeoff between faithfulness and interpretability controllable by the practitioners.

**Actions Required To Increase Overall Recommendation:**

Further explain the technical contribution and to distinguish from directly applying related work.

**Clarity:**





**Overall Review:**

Overall the idea of this paper is quite interesting. The authors first observe the feature gap that hinders wider adoption of the existing HPO tools. Then, the authors propose simple yet effective approaches to provide interpretability without infeasible overhead. Further, the authors do an extensive evaluation to illustrate the effectiveness of their approach, namely how well the symbolic models provide explanation and meanwhile accurately to capture the dynamics. The downside of this work is that it is unclear how much contribution is original to the authors as opposed to just applying and trying a combination of existing algorithms and tune the approach to work for certain settings.

**Potential Impact On The Field Of Automl:**

Overall the idea of this paper is interesting and addresses a timely problem. Although many existing tools can find a well-performing configuration quite efficiently, a major blocker for them to be widely adopted is the lack of insights into the optimization landscape as these tools mostly work in a blackbox fashion. The proposed method allows the practitioner to gain more insight on the HPO process which is important and require further research in the field of AutoML.

**Review Confidence:**

3: You are fairly confident in your assessment. It is possible that you did not understand some parts of the submission or that you are unfamiliar with some pieces of related work.

**Review Rating:**

7: Weak Accept: Technically sound paper with moderate-to-high impact and strong evaluation, with perhaps some minor flaws.

**Review Summary:**

Simple yet effective approach. The paper is well written. Clarification is needed to further explain the technical contribution and to distinguish from directly applying related work.

**Technical Quality And Correctness:**

Overall, the technical quality of this work is solid. Some statements by the authors are not fully explained but they are not likely to affect the conclusions. For example, in Sec 4.1, the authors mention that gradient-based methods work less reliably compared to traditional genetic programming works. It would be better to explain why because it does not seem intuitive enough to skip. Similarly, more theoretical analysis on the impact to the performance of using surrogate model to estimate the randomly selected points rather than actually evaluating them could further enhance the technical quality of this work.

---

> ### Author Response · Authors · 2023-04-26
> **Response to Reviewer PYEc**
>
> Thank you very much for your comments and feedback. Below, we provide responses to each of the specific comments you raised in your review. We hope that our responses adequately address your concerns, and are happy to answer any follow-up questions.
>
> > For example, in Sec 4.1, the authors mention that gradient-based methods work less reliably compared to traditional genetic programming works. It would be better to explain why because it does not seem intuitive enough to skip.
>
> The gradient-based methods were brittle in some preliminary experiments we did, meaning that in quite a few cases, we could not recover simple functions, such as polynomial functions, with the approach. Thank you for pointing this out - we adapted the corresponding part of the paper to be more precise.
>
> > The downside of this work is that it is unclear how much contribution is original to the authors as opposed to just applying and trying a combination of existing algorithms and tune the approach to work for certain settings.
>
> While our proposed workflow leverages well-known, existing algorithms, we argue that our contributions are far beyond the simple combination of existing algorithms. Concretely, we show that learning a symbolic model naively on metadata collected during the HPO process does not yield a good explanation, due to bias caused by the HPO sampling strategy, and propose a procedure to alleviate this problem. Furthermore, we propose a systematic approach to control the trade-off between the symbolic model's faithfulness and interpretability based on a parsimony hyperparameter. In addition to the previous contributions, we now added a section (Section 4.1.3.) in which we propose steps to extend the workflow such that useful explanations can be obtained when optimizing multiple hyperparameters, for which we will add an example within the next days.
>
> We will inform you once the open points are added to the paper. Thank you again for your valuable input.

---

### Official Review · Reviewer_bYyg · 2023-04-12

**Potential Impact On The Field Of Automl Rating:** 4
**Technical Quality And Correctness Rating:** 2
**Clarity Rating:** 3

**Summary Of Contributions:**

The paper focuses on introducing explanability and interpretability to the very critical but opaque hyperparameter optimization or HPO process. To this end, the paper utilizes the symbolic regression framework to model the hyperparameter-loss relationship utilizing the meta-data obtained from any HPO execution. The paper highlights the common tradeoff in symbolic regression between expression fidelity (the ability of learned expression to fit the data) and the expression complexity. Furthermore, the paper highlights a challenge in applying the symbolic regression framework to meta-data from a Bayesian Optimization based HPO scheme, and presents the advantages and disadvantages of two ways to mitigating this challenge.


**Actions Required To Increase Overall Recommendation:**

I am happy to increase my score if the authors (or other reviewers) can point out how my concerns regarding the fundamental issues with the proposed idea is not justified or if it is a result of my misunderstanding of the idea or results.


**Clarity:**

In terms of clarity, this paper is in a great position. The authors do a good job at motivating the problem and the use of symbolic regression. The paper identifies and discusses various limitations.

Beyond the technical aspects discussed in the **Technical Quality and Correctness** section, one thing that is not clear is regarding the SR(BO-GP) variant -- If we are modeling the surrogate function, how are the predictions for the random samples selected for regions with high uncertainty? Are we just selecting the mean of the posterior or are we sampling from the posterior distribution? How would the results be affected, if at all, by these different choices.

Moreover, one can argue that the SR expression for the surrogate model will probably be evolving through the HPO -- how does that play into the explanation?


**Overall Review:**

As I have detailed in the **Technical Quality and Correctness** section, the main strengths of this paper are the well-motivated problem and a natural, intuitive solution.

However, as detailed in the same section, one of the main concerns with the proposed idea is that it is not clear how much interpretability a simple expression for the (surrogate or actual) loss surface brings to the HPO process given the interpretability-fidelity tradeoff and the instability of the interpreted dependencies.
The choice of small HPO problems (problems with just 2 hyperparameters) does not appropriately highlight the advantage of utilizing symbolic regression over existing explanability schemes based on partial dependency plots.


**Potential Impact On The Field Of Automl:**

Hyperparameter optimization is a well recognised tool in machine learning for obtaining critical performance improvements. A mechanism to make this process explainable and interpretable will have a wide ranging impact in the area of machine learning, for easier adoption to potentially better understanding of theoretical properties of models and hyperparameters. This paper studies one such mechanism and thus can be very important.


**Review Confidence:**

4: You are confident in your assessment, but not absolutely certain. It is unlikely, but not impossible, that you did not understand some parts of the submission or that you are unfamiliar with some pieces of related work.

**Review Rating:**

7: Weak Accept: Technically sound paper with moderate-to-high impact and strong evaluation, with perhaps some minor flaws.

**Review Summary:**

Based on the above review, the proposed idea seems an interesting direction, but in my opinion, the current conclusions and explanations seem limited or not well-supported. Hence, I recommend a rejection.


**Technical Quality And Correctness:**

The paper considers a very natural choice of symbolic regression to model the hyperparameter-loss relationship to provide insights into the HPO process. The proposed idea is evaluated on various HPO problems with large number of trials, generating results with statistically significant confidence intervals.

The paper also clearly highlights various limitations of the proposed approach and the evaluation. This is very critical because it highlights that the authors have viewed the problem from various aspects, identifying potential challenges and mitigation.

There are some technical concerns that I will list in detail here:


One main concern is that it seems that we might be conflating the use of an interpretable modeling of the hyperparameter-loss relationship with the interpretability or explainability of the HPO process. Even if we assume that we are able to obtain a perfectly faithful yet very low-complexity expression with Symbolic Regression (SR), this expression highlights what loss function the HPO process is **estimating** and optimizing for. It is not clear how this leads to an answer of the critical questions of "why were these hyperparameters selected".

This issue gets more complicated when we consider the parsimony hyperparameter. It is not clear how the faithfulness or fidelity of the SR model plays into the explanation. As the complexity of the SR-induced expression increases, the interpretability decreases while the fidelity improves. In this case, as we span the tradeoff spectrum, it would seem that the explanations (as obtained via the SR-induced expression) themselves would be changing, which is counter-intuitive. How can we address the fact that the SR model does not really match the true loss landscape in our explanations?

This issue is also related to the stability issue in symbolic regression -- if different runs produce different expressions (of potentially same level of fidelity), then it is not clear which one is the right one. This issue is highlighted in the results of this paper itself -- for the same problem (hyperparameters and dataset), there are two possible relationships that SR finds; and one of them does not depend on one of the hyperparameters `batch_size` at all. This is a significant level of instability in the explanations. Furthermore, $\exp(\text{alpha})$ and $\sqrt{|\sin \text{alpha}|}$ are very different functions -- $\exp(\text{alpha})$ can grow forever with $\text{alpha}$, while $\sqrt{|\sin \text{alpha}|}$ is bounded with a periodicity.

Another (relatively smaller) concern is the use of HPO problems with just 2 hyperparameters. The PDP based schemes are mentioned to be only limited to studying the effect of 2 or less hyperparameters at a time. The proposed SR based scheme should be relatively agnostic to the number of hyperparameters -- this aspect of the proposed scheme should have been highlighted in the empirical evaluations with HPO problems with a large number of hyperparameters.

---

> ### Author Response · Authors · 2023-04-26
> **Response to Reviewer bYyg, Part 1**
>
> Thank you very much for your comments and feedback. Below, we provide responses to each of the specific comments you raised in your review. We hope that our responses adequately address your concerns, and are happy to answer any follow-up questions.
>
>  > One main concern is that it seems that we might be conflating the use of an interpretable modeling of the hyperparameter-loss relationship with the interpretability or explainability of the HPO process. Even if we assume that we are able to obtain a perfectly faithful yet very low-complexity expression with Symbolic Regression (SR), this expression highlights what loss function the HPO process is estimating and optimizing for. It is not clear how this leads to an answer of the critical questions of "why were these hyperparameters selected".
>
> First of all, we would like to clarify that we do not claim that our approach answers the question of why certain hyperparameters or hyperparameter configurations were selected. However, as we detail in Section 4.2, we envision several ways how the symbolic expression can improve the understanding of the HPO process. In particular, our approach allows (i) to characterize the HPO loss landscape mathematically through a closed-form expression and (ii) allows one to plot it. We believe that these are relevant insights into the problem gained through our approach.
>
> > This issue gets more complicated when we consider the parsimony hyperparameter. It is not clear how the faithfulness or fidelity of the SR model plays into the explanation. As the complexity of the SR-induced expression increases, the interpretability decreases while the fidelity improves. In this case, as we span the tradeoff spectrum, it would seem that the explanations (as obtained via the SR-induced expression) themselves would be changing, which is counter-intuitive. How can we address the fact that the SR model does not really match the true loss landscape in our explanations?
>
> As Figure 5 shows, the SR model trained based on the randomly sampled points evaluated by the GP model performs quite well even for a rather low complexity. Consider, for example, the XGBoost example on the blood-transfusion-service-center dataset, where we already achieve a very faithful expression with an operation count of 1-3. This coincides with previous findings on the HPO loss landscape, which indicate that HPO loss landscapes are rather benign [1] and thus, it should be feasible to model them with a low complexity of an SR model. There are certainly learner and dataset combinations where we need a higher complexity to achieve a very faithful model, as, for example, in the SVM case on the Australian dataset. However, even in that case, we can achieve a reasonable faithfulness with still a small operation count of 4-5. Correspondingly, there is no apparent need to address that the SR model will not perfectly match the true loss landscape, from our point of view, as one apparently achieves a reasonable faithfulness with a high default parsimony value in practice. Figure 5 only highlights a simple way to fine-tune the parsimony for those users who are not satisfied by the default setting.

---

> > ### Author Response · Authors · 2023-04-26
> > **Response to Reviewer bYyg, Part 2**
> >
> > > This issue is also related to the stability issue in symbolic regression -- if different runs produce different expressions (of potentially same level of fidelity), then it is not clear which one is the right one. This issue is highlighted in the results of this paper itself -- for the same problem (hyperparameters and dataset), there are two possible relationships that SR finds; and one of them does not depend on one of the hyperparameters batch_size at all. This is a significant level of instability in the explanations. Furthermore, $\exp(\text{alpha})$ and $\sqrt{|\sin\text{alpha}|}$ are very different functions -- $\exp(\text{alpha})$  can grow forever with alpha, while $\sqrt{|\sin\text{alpha}|}$ is bounded with a periodicity.
> >
> > Concerning the stability issue in symbolic regression: There often are various explanations for a specific behavior, e.g. a multitude of different equations in a class of functions giving about the same minimum error rate in a bounded space. This is a general issue of many explanation techniques, called the Rashomon effect [2, 6], and thus also applies to the technique proposed in our paper.
> > Nevertheless, we argue that leveraging symbolic regression, the user can obtain a reasonable number of explanations and leverage these as a usable hypothesis space of explanations for further studies.
> >
> > Furthermore, one could make use of conformal prediction for quantifying the uncertainty of the different expressions [3]. We added this as possible future work in Section 6.
> >
> > Concerning the concrete functions shown in the paper: After re-checking the functions, we actually spotted a typo in the first function, which we corrected now. While indeed it is correct that $\exp(\text{alpha})$ and $\sqrt{|\sin\text{alpha}|}$ are different functions with different behaviors for growing alpha, it needs to be considered that the resulting functions only describe the functional relationship within the bounds that were defined for the hyperparameters. Within these bounds, the relationship described by the two functions is actually quite similar. Notably, the second function does not depend on the batch size at all, indicating that alpha is more important than the batch size. In Section 5.2, we added a plot showing the importance values for both hyperparameters calculated using functional ANOVA [5], indicating that alpha is indeed by far more important than the batch size. This is quite in line with recent findings [8] and shows a useful degree of faithfulness of our approach.
> >
> > > Another (relatively smaller) concern is the use of HPO problems with just 2 hyperparameters. The PDP based schemes are mentioned to be only limited to studying the effect of 2 or less hyperparameters at a time. The proposed SR based scheme should be relatively agnostic to the number of hyperparameters -- this aspect of the proposed scheme should have been highlighted in the empirical evaluations with HPO problems with a large number of hyperparameters.
> >
> > We agree that the proposed approach should be agnostic to the number of hyperparameters. We decided to consider a small number of hyperparameters in our explanations mainly for two reasons:
> > 1) Previous work [4, 5] has shown that most performance variation is attributable to just a few hyperparameters.
> > 2) Interpretability decreases with the number of hyperparameters involved in the resulting formula. Furthermore, with more than two hyperparameters, plotting the formula becomes more complex or impossible.
> >
> > In Section 4.1.3, we have added a more detailed recommendation on how to obtain useful explanations when optimizing multiple hyperparameters, and will add an example for this within the next days.
> >
> > > One thing that is not clear is regarding the SR(BO-GP) variant -- If we are modeling the surrogate function, how are the predictions for the random samples selected for regions with high uncertainty? Are we just selecting the mean of the posterior or are we sampling from the posterior distribution? How would the results be affected, if at all, by these different choices.
> >
> > Thank you for pointing out this lack of clarity in our formulation. We clarified this in the paper: To obtain the predictions for the random samples, we leverage the surrogate model’s posterior mean.
> > Given the favorable results obtained using this method, we did not attempt to sample from the posterior distribution instead. However, we assume that the additional noise introduced by this might not be helpful to learn a low-complexity SR model. Leveraging the mean of the predictive distribution can be seen as a form of label smoothing [7] in order to stabilize the learning process. We hypothesize that sampling from the posterior would lead to even more complex and varying explanations.

---

> > > ### Author Response · Authors · 2023-04-26
> > > **Response to Reviewer bYyg, Part 3**
> > >
> > > > Moreover, one can argue that the SR expression for the surrogate model will probably be evolving through the HPO -- how does that play into the explanation?
> > >
> > > At the moment, it does not play into the explanation at all, as we do a posthoc analysis with all available data at the end. We could also compute intermediate explanations, which would most likely be more noisy in the beginning as the meta-dataset is smaller. However, although this is an interesting idea, it is out of scope of this paper.
> > >
> > > We will inform you once the open points are added to the paper. Thank you again for your valuable input.
> > >
> > > [1] Pushak, Yasha, and Holger Hoos. "AutoML Loss Landscapes." ACM Transactions on Evolutionary Learning 2.3 (2022): 1-30.
> > > [2] Breiman, Leo. “Statistical modeling: The two cultures.” Quality Engineering 48 (2001): 81-82.
> > > [3] Angelopoulos, Anastasios Nikolas and Stephen Bates. “A Gentle Introduction to Conformal Prediction and Distribution-Free Uncertainty Quantification.” ArXiv abs/2107.07511 (2021).
> > > [4] Rijn, Jan N. van and Frank Hutter. “Hyperparameter Importance Across Datasets.” Proceedings of the 24th ACM SIGKDD International Conference on Knowledge Discovery & Data Mining (2017).
> > > [5] Hutter, Frank et al. “An Efficient Approach for Assessing Hyperparameter Importance.” International Conference on Machine Learning (2014).
> > > [6] Leventi-Peetz, Anastasia-M., and Kai Weber. "Rashomon Effect and Consistency in Explainable Artificial Intelligence (XAI)." Proceedings of the Future Technologies Conference (FTC) 2022, Volume 1. Cham: Springer International Publishing, 2022.
> > > [7] Müller, Rafael, Simon Kornblith, and Geoffrey E. Hinton. "When does label smoothing help?." Advances in neural information processing systems 32 (2019).
> > > [8] Godbole, Varun et al. (2023). Deep Learning Tuning Playbook. Version 1.

---

> > > > ### Comment · Reviewer_bYyg · 2023-05-01
> > > > **Thank you for detailed thorough response**
> > > >
> > > > I thank the authors for responding to my comments in detail. The authors raise a lot of good counterpoints. For example, I appreciate the explanation regarding why the two quite different symbolic expressions might be similar is a particular range of value which of interest in the hyperparameter optimization. I also appreciate the additional clarifications and the discussion and new experiments on hyperparameter importances.
> > > >
> > > > I am happy to increase the score. However, I still maintain the following reservation.
> > > > We are still just modeling the surrogate function with a symbolic expression and then using this expression as an "explanation for hyperparameter optimization". Furthermore, based on the hyperparameter importance discussion, we are also just focusing on modeling the surrogate model with a symbolic expression of the hyperparameters we consider the most important. I cannot speak for how explainable partial dependency plots of the surrogate function are either, but in my opinion, I am not sure if such a symbolic expression as an explanation is providing any insights about the main problem since we are modeling the surrogate function, not the real one. And it is known that the accuracy of the surrogate model is fairly variable across the whole hyperparameter space. The symbolic regression is not taking that into account so it is not clear how faithful the symbolic regression is to the true function, and not just having low RMSE on the data generated from the surrogate model. If we are modeling the wrong thing, a succinct clean expression is not going to be the right explanation.

---

### Official Review · Reviewer_8gQo · 2023-04-12

**Potential Impact On The Field Of Automl Rating:** 3
**Technical Quality And Correctness Rating:** 4
**Clarity Rating:** 3
**Actions Required To Increase Overall Recommendation:** See review summary.

**Summary Of Contributions:**

The paper suggests an approach to make the black-box HPO process more transparent and explainable. In order to do so, the meta-data generated by a Bayesian Optimization based HPO tool (SMAC) is used as training data for a symbolic regressor. The goal of using symbolic regression is to learn the relationship between the hyperparameters and the loss value. By looking at the explicit formula discovered by symbolic regression, insights about the loss landscape of the HPO problem can be gained. However, the meta-data generated by BO-HPO tools is biased towards certain regions of the loss landscape. To avoid this, the meta-data of randomly sampled configurations is used by leveraging the Gaussian Process model learned during the BO process to approximate the loss values of randomly sampled configurations. Experimental evaluations of the HP-Bench benchmark show this approach to be favorable.

**Clarity:**

The work is clear in most parts. While Figures 1 and 2 foster the understanding of the problem and the workflow, the added value from Figure 3b is limited. The sampling bias of BO is already mentioned in the text and should not require a separate figure. Also, the formulas presented in Section 5.2, which are the main results of this approach are not explained any further.


**Overall Review:**

Pros:

1. The idea to use symbolic regression for better explainability of the HPO process is a computationally cheap method to generate inisghts and could thus be widely adopted.
2. The proposed workflow is technically sound and straight-forward to apply.
3. Evaluations on the HPO-Bench show the approach to be able to accurately approximate the true cost values, thus the methods seems to working well.

Cons (Major):

1. The GP baseline in Table 1 is the only baseline that the approach is compared to. What about other “simple” regression techniques? Is it obvious that linear regression or logistic regression (that are both interpretable in some sense as well) would produce worse results?
2. The main results of the experiments, namely the formulas in Section 5.2 are not explained at all. How can these formulas help practitioners of HPO to understand the loss landscape better? What do the factors (e.g. 0.078) mean?

Cons (Minor):

1. In the experimental evaluations just two hyperparameters for each type of machine learning model are analyzed. This is a very small subset of available (numerical) hyperparameters

**Potential Impact On The Field Of Automl:**

The proposed approach could help users of HPO methods to get a better understanding of the loss landscape of the HPO problem they are looking to tackle. As HPO methods based on BO work as black-box, generating insights from this that can be analyzed by a human is important for the general field of AutoML. However, similar approaches exist already in the literature, as also mentioned by the authors (e.g. PDP plots).


**Review Confidence:**

4: You are confident in your assessment, but not absolutely certain. It is unlikely, but not impossible, that you did not understand some parts of the submission or that you are unfamiliar with some pieces of related work.

**Review Rating:**

7: Weak Accept: Technically sound paper with moderate-to-high impact and strong evaluation, with perhaps some minor flaws.

**Review Summary:**

Overall, the idea and the proposed workflow with symbolic regression for explainability seem to be novel and well-suited for the problem. But adding comparisons to additional baselines is necessary to assess its real impact and a more detailed explanation of the final results is absolutely crucial for practitioners to use this approach.


**Technical Quality And Correctness:**

The approach is technically sound and correct. The claims are supported by experimental evaluations. The experimental setup is described in the appendix.

---

> ### Author Response · Authors · 2023-04-26
> **Response to Reviewer 8gQo**
>
> Thank you very much for your comments and feedback. Below, we provide responses to each of the specific comments you raised in your review. We hope that our responses adequately address your concerns, and are happy to answer any follow-up questions.
>
> > While Figures 1 and 2 foster the understanding of the problem and the workflow, the added value from Figure 3b is limited. The sampling bias of BO is already mentioned in the text and should not require a separate figure.
>
> The insight is indeed limited for those who are aware how prominent the sampling bias is. However, we believe that those readers who are not too aware of the problem might find this a useful visualization of the concept.
>
> > The GP baseline in Table 1 is the only baseline that the approach is compared to. What about other “simple” regression techniques? Is it obvious that linear regression or logistic regression (that are both interpretable in some sense as well) would produce worse results?
>
> Thank you for the idea to compare against linear regression as an alternative to symbolic regression. We will add this as additional baseline in the paper within the next days. However, in view of the landscape analysis by Pushak and Hoos [1], we do not believe that linear regression will be a good fit in most cases. Furthermore, we believe that logistic regression is not a good fit for our case, as it estimates the odds of an event while we are interested in modeling the HPO loss landscape as a regression problem.
>
> > The main results of the experiments, namely the formulas in Section 5.2 are not explained at all. How can these formulas help practitioners of HPO to understand the loss landscape better? What do the factors (e.g. 0.078) mean?
>
> We agree that some more explanation on the resulting formulas would be valuable for the reader and added an explanation in Section 5.2.
>
> > In the experimental evaluations just two hyperparameters for each type of machine learning model are analyzed. This is a very small subset of available (numerical) hyperparameters.
>
> In general, the proposed approach should be agnostic to the number of hyperparameters. Nevertheless, we decided to consider a small number of hyperparameters in our explanations mainly for two reasons:
> 1) Previous work [2, 3] has shown that most performance variation is attributable to just a few hyperparameters.
> 2) Interpretability decreases with the number of hyperparameters involved in the resulting formula. Furthermore, with more than two hyperparameters, plotting the formula becomes more complex or impossible.
>
> In Section 4.1.3, we added a more detailed recommendation on how to obtain useful explanations when optimizing multiple hyperparameters, and will add an example for this within the next days.
>
> We will inform you once the open points are added to the paper. Thank you again for your valuable input.
>
> [1] Pushak, Yasha, and Holger Hoos. "AutoML Loss Landscapes." ACM Transactions on Evolutionary Learning 2.3 (2022): 1-30.
> [2] Rijn, Jan N. van and Frank Hutter. “Hyperparameter Importance Across Datasets.” Proceedings of the 24th ACM SIGKDD International Conference on Knowledge Discovery & Data Mining (2017).
> [3] Hutter, Frank et al. “An Efficient Approach for Assessing Hyperparameter Importance.” International Conference on Machine Learning (2014).

---

> > ### Comment · Reviewer_8gQo · 2023-04-30
> > **Response to Authors**
> >
> > I thank the authors for addressing my concerns and running additional experiments. Regarding one of my main concerns: As the authors point out correctly in the current manuscript version, linear regression indeed does not match the SR performance consistently (though it is sometimes close).
> >
> > I also appreciate the idea of adding an ANOVA to the workflow to include a larger number of hyperparameters in the analysis, but do have one more question regarding this new addition: Are there any new limitations that arise from the usage of an ANOVA in connection with SR?

---

> > > ### Author Response · Authors · 2023-05-01
> > > **Response to Reviewer 8gQo**
> > >
> > > The limitations of fANOVA itself apply in principle. In particular:
> > >
> > > 1) fANOVA can yield misleading results when dealing with correlated features due to extrapolation to unlikely combinations of feature values (i.e., hyperparameter values, in our case) [1]. However, in contrast to the standard setting of interpreting machine learning models, there is no inherent data distribution on the hyperparameter configuration space in the context of HPO. Thus, this limitation is no concern for the HPO setting.
> > >
> > > 2) When interaction effects are present, the main effects obtained by fANOVA can be misleading, as they capture the effect of varying the hyperparameter averaging across all instantiations of all other hyperparameters [2]. However, recent findings by Pushak and Hoos [3] suggest that there are few strong interactions in HPO problems, which makes this limitations less of a concern in practice.
> > >
> > > [1] G. Hooker. Generalized functional anova diagnostics for high-dimensional functions of dependent variables. Journal of Computational and Graphical Statistics, 2007.
> > > [2] Hutter, Frank et al. “An Efficient Approach for Assessing Hyperparameter Importance.” International Conference on Machine Learning (2014).
> > > [3] Pushak, Yasha, and Holger Hoos. "AutoML Loss Landscapes." ACM Transactions on Evolutionary Learning 2.3 (2022): 1-30.

---

> > > > ### Comment · Reviewer_8gQo · 2023-05-01
> > > > **Response to Authors**
> > > >
> > > > I thank the authors for the quick clarification - it would be helpful to include this information in the paper (as part of the appendix) as well.
> > > >
> > > > As all my concerns have been addressed I am raising my score and now recommend the acceptance of the paper.

---

> > > > > ### Author Response · Authors · 2023-05-02
> > > > > **Response to Reviewer 8gQo**
> > > > >
> > > > > Thank you for your feedback. As you suggested, we added the information on the limitations of fANOVA in the appendix.

---

### Official Review · Reviewer_7csY · 2023-04-13

**Potential Impact On The Field Of Automl Rating:** 4
**Technical Quality And Correctness Rating:** 4
**Clarity Rating:** 4
**Actions Required To Increase Overall Recommendation:** No specific suggestion

**Summary Of Contributions:**

While existing hyperparameter optimization (HPO) methods efficiently identify well-performing hyperparameter configurations, they often lack transparency and insights.  In this paper, the authors propose a  method that uses symbolic regression on meta-data obtained through Bayesian optimization (BO) during HPO. Symbolic regression allows them to get explicit formulas that quantify the relationship between hyperparameter values and model performance. This approach makes the HPO process more explainable and human-centered.

Since symbolic regression directly to meta-data collected during HPO can be influenced by the sampling bias introduced by BO. The authors propose to fit the symbolic regression on the surrogate model trained during BO, which can approximate the true underlying loss landscape.


**Clarity:**

The paper is well-written. The description of the different stages of the method is clear. Moreover, appendices contain very interesting materials that assess the contribution and its potential impact.


**Overall Review:**

Pro

The state of the art is meticulously presented, offering a comprehensive overview of relevant literature in the field.

Despite being extensive and technical, the background information is clearly articulated, making it accessible to a wide range of readers.

The proposed method is intriguing and offers several opportunities for future refinement and improvement.

The limitations of the approach are transparently highlighted, such as its focus only on numerical parameters.

The experiments use an up-to-date tool (SMAC), which enhances the reliability and validity of the study.

A critical issue, the trade-off between faithfulness and interpretability, is thoroughly explored, providing a thought-provoking discussion on this topic.


Cons

A more detailed description of the GP part is missing to my opinion

Considerations on landscapes analysis are still limited


**Potential Impact On The Field Of Automl:**

Different types of users can use such an approach, including data scientists and machine learning practitioners who can benefit from insights into the HPO process to make informed decisions. It can also be helpful to better understand the HPO loss landscape by inspecting the properties of the obtained formulas.

**Review Confidence:**

4: You are confident in your assessment, but not absolutely certain. It is unlikely, but not impossible, that you did not understand some parts of the submission or that you are unfamiliar with some pieces of related work.

**Review Rating:**

8: Accept: Technically sound paper with major impact and strong evaluation, with perhaps some minor flaws.

**Review Summary:**

The main purpose of the paper was to make the HPO process more human-centered by proposing a practical workflow for practitioners to gain insights into the underlying HPO problem. The proposed approach uses symbolic regression to derive a closed-form expression that captures the relationship between hyperparameter values and their performance in an interpretable manner. The authors point out that simply learning a symbolic model on meta-data collected during the HPO process can be biased. To address this issue, they have proposed learning on randomly sampled configurations whose performance is approximated by the HPO process' surrogate model. The experimental evaluation demonstrates that a balance between the interpretability and faithfulness of the symbolic model can be achieved by systematically setting a parsimony hyperparameter.

This is clearly a relevant and good-quality contribution to AutoML


**Technical Quality And Correctness:**

Experiments have been conducted on a set of representative classic benchmarks. Experimental conditions and processes are sufficiently described. The technical descriptions of the different stages of the methods are detailed.

---

> ### Author Response · Authors · 2023-04-26
> **Response to Reviewer 7csY**
>
> Thank you very much for your comments and feedback.
>
> We have carefully considered your comments, and would like to clarify one point you raised. Concerning your comment “A more detailed description of the GP part is missing”: Are you referring to the part on Bayesian optimization in the background section (Section 3), or on how we use the GP surrogate to generate predictions for the randomly sampled points? Concerning the background section, we are referring to the book on GPs by Rasmussen, which readers, who are not too familiar with the topic can, can skim. Concerning the latter, we added a comment in Section 5 to clarify that to obtain the predictions for the random samples, we leverage the surrogate model’s posterior mean.
>
> Thank you again for your valuable input.

---

### Review · Reproducibility_Reviewer_SJZo · 2023-04-16

**Completeness Of Code And Dataset Supplement Rating:** 4
**Usability And Ease Of Reproducibility Rating:** 4

**Actions Required To Increase The Reproducibility And Overall Recommendation:**

Please make any changes requiring minimal effort which would improve the experience of the user with only the computational budget to run some subset of your results.

**Completeness Of Code And Dataset Supplement:**

My understanding is that:
- Figures 1/2 are illustrations of the procedure, probably created by hand.
- Figure 3 is produced by plot_2d_hpobench.py
- Figure 4 is produced by plot_learning_curves_hpobench.py
- Figure 5 is produced by plot_complexity_vs_rmse.py
- Table 1 is produced by metrics_hpobench.py

Furthermore, it seemed like running the scripts automatically downloaded the necessary dataset.

Therefore, everything is accounted for.

**Overall Reproducibility Review:**

The README was to the point, the directory well-structured, and the python environment well-configured.

**Review Confidence:**

3: You are fairly confident in your assessment. It is possible that you did not understand some parts of the submission or that you are unfamiliar with some pieces of the code or data.

**Review Rating:**

9: Strong Accept, all aspects of this are easily reproducible.

**Review Summary:**

This repository was very easy to set up and get running. Furthermore, running the scripts results in extensive logging, which can be helpful for trying to identify any issues that arise.

I was able to run the scripts for the actual computation seemingly without issue. I only tried them for a single dataset. I was not able to get all of the plotting scripts to work, but I suspect this is because I ran only a subset of the computing scripts. Ideally, the repo would be set up so as to allow an interested but casual user to run only a subset of the scripts. But overall this repository was well organized and I review it highly.

**Summary Of Necessary Code And Dataset Supplement:**

This article essentially proposes a post-hoc explanation of a hyperparameter optimization procedure: first we conduct a hyperparameter optimization (SMAC in their case), and then perform symbolic regression (via a genetic algo) to try to explain the sensitivity.

Therefore, there are essentially two main computational components, run in succession.

The authors use the "HPOBench" benchmark to evaluate their methodology.

**Usability And Ease Of Reproducibility:**

The README file is great, even including a "TLDR" section. I had no trouble creating their python environment, and the scripts for running the comparisons ran on the first try.

I only tried running the first of the 40 datasets they considered, which was also the default in the README file.

However, I think there were a few issues specifically related to the fact that I only ran some subset of the datasets. The scripts "plot_learning_curves_hpobench.py" and "plot_complexity_vs_rmse.py" reported errors due to not being able to find certain files, and seemed not to produce output.

A less serious issue is that it seemed to me like a lot of unnecessary overhead was being done for the special case of only trying 1 of the 40 datasets (e.g. I believe other datasets were being downloaded or validated or something like that based on the log output). Someone interested in quickly reproducing just some subset of the results would be inconvenienced.

---

### Author Response · Authors · 2023-04-26
**Overall Response**

Dear reviewers,

thank you for your valuable feedback and questions, which we answer below in comments to each reviewer. We are currently running additional experiments with another baseline to the symbolic regression model. Furthermore, we are running experiments on how to deal with problems featuring more than two hyperparameters to optimize, along the lines of the additional workflow element in Section 4.1.3. We will alert you again, once these additional experimental results are included in the paper. However, we have already integrated all requested changes not requiring additional experiments in the current form of the manuscript. For convenience, these changes are highlighted in red.

---

> ### Author Response · Authors · 2023-04-28
> **Follow-up Response**
>
> Dear reviewers,
>
> we would like to follow up on our recent response to your comments on our paper. If you have any further concerns or questions, we would greatly appreciate it if you could let us know at your earliest convenience. We want to ensure that we have addressed all of the questions that you raised. Thank you for your time and consideration.

---

> > ### Author Response · Authors · 2023-04-29
> > **Additional Experiments Added**
> >
> > Dear reviewers,
> >
> > as announced in our previous response, we ran additional experiments with another baseline to the symbolic regression model. We now added the results to Table 1 in Section 5.1.
> > Furthermore, we provided an example on how to deal with problems featuring more than two hyperparameters to optimize in Section 5.2.